# Temperature-Dependent Polymorphism and Phase Transformation of Friction Transferred PLLA Thin Films

**DOI:** 10.3390/polym14235300

**Published:** 2022-12-04

**Authors:** Jinghua Wu, Xing Chen, Jian Hu, Shouke Yan, Jianming Zhang

**Affiliations:** 1Key Laboratory of Rubber-Plastics, Ministry of Education/Shandong Provincial Key Laboratory of Rubber-Plastics, Qingdao University of Science & Technology, Qingdao 266042, China; 2State Key Laboratory of Chemical Resource Engineering, Beijing University of Chemical Technology, Beijing 100029, China

**Keywords:** highly oriented PLLA film, friction transfer technique, phase transition, β-form

## Abstract

Poly(L-lactic acid) (PLLA) thin films with a highly oriented structure, successfully prepared by a fast friction transfer technique, were investigated mainly on the basis of synchrotron radiation wide-angle X-ray diffraction (WAXD) and Fourier transform infrared spectroscopy (FTIR). The crystalline structure of the highly oriented PLLA film was remarkably affected by friction transfer temperatures, which exhibited various crystal forms in different friction temperature regions. Interestingly, metastable β-form was generated at all friction transfer temperatures (70–140 °C) between T_g_ and T_m_, indicating that fast friction transfer rate was propitious to the formation of β-form. Furthermore, the relative content among β-, α′-, and α-forms at different friction temperatures was estimated by WAXD as well as FTIR spectroscopy. In situ temperature-dependent WAXD was applied to reveal the complicated phase transition behavior of PLLA at a friction transfer temperature of 100 °C. The results illustrated that the contents of β- and α′-forms decreased in turn, whereas the α-form increased in content due to partially melt-recrystallization or crystal perfection. Moreover, by immersing into a solvent of acetone, β-, α′-form were transformed into stable α-crystalline form directly as a consequence. The highly oriented structure was maintained with the chain perfectly parallel to friction transfer direction after acetone treatment, evidenced by polarized FTIR and polarized optical microscopy (POM) measurements.

## 1. Introduction

Poly(L-lactic acid) (PLLA), as a promising bio-based plastic that may be helpful in solving the environmental pollution problem and lessen the dependence on petroleum-based plastics, has drawn much attention recently [1,2]. Due to the advantages of mechanical property and easy processability, PLLA has been widely used in packaging and biomedical fields for substitution of traditional commodity plastic [3,4,5,6,7,8,9]. For the purpose of expanding the end-uses, it is of great importance to control the mechanical properties intimately related to the crystalline structure and morphology [10,11,12]. As a semi-crystalline polymer, various crystal modifications of PLLA, including α-form, α′- (or δ-) form, β-form, γ-form, and stereocomplex between PLLA and Poly(D-lactic acid) (PDLA), can be obtained by the regulation of crystal condition [13,14,15,16,17,18,19,20]. α-form is the most stable modification and adopts a 10/3 helical conformation in the orthogonal unit cell, which can be directly formed at high temperature (>120 °C) or from solution crystallization [21,22,23,24,25]. Independent from the α-form, α′- (or δ-) form with an up–down disordered helix shows more disordered multi-domain structure and is generated at relatively low temperature (<100 °C) [19,26]. The phase transition behavior from disordered α′- to ordered α-form concerning the structure, kinetics, and thermodynamic properties have been extensively investigated by many groups [15,19,27,28,29,30,31,32]. Furthermore, the PLLA β-form is a metastable, frustrated structure that adopts a 3/1 helical conformation obtained by stretching the α-form at high temperature [33], solution-spinning [34], solid-state coextrusion [35], under pressure and shear [36], single crystal growth from designated solutions [37], at a low crystallization temperature (75–100 °C), or from PLLA/PDLA blend [38] or supramolecular PLLA bonded by multiple hydrogen bonds [39]. The refinement structure of β-form proposed by Tashiro et al. reveals that it contains six helical chains in one rectangular unit cell of P1 space group with alternative upward and downward chain [40]. In addition, they proposed the stress-induced phase transition mechanism α-form with 10/3 helices to β-form with 3/1 helical conformation via α′- (δ-) form under deformation at different temperatures. In our recent study, it was found that β-form fibrils reorganized and perfected upon further annealing by utilizing orientated PLLA ultrathin melt-draw films [41].

On the other hand, orientation is an efficacious approach to obtaining polymeric materials with excellent properties, and the orientation structures arise from the arrangement of the molecule chain, from a coil to an extended chain [42]. Various methods have been developed to extend the polymer chain, such as uniaxial drawing and biaxial drawing [43,44]. Blowing molding has been used to manufacture high-performance polymer products. Newly developed methods such as the friction transfer technique, dip-coating technique, the Langmuir–Blodgett (LB) technique, and liquid crystalline (LC) self-organization were used to arrange the molecular chain in the lab to obtain high physical performance [45,46,47,48,49,50,51]. The method of friction transfer was first proposed by Wittman and Smith in 1991 to prepare the highly oriented poly (tetrafluoroethylene) (PTFE) film [52]. Compared to other methods, friction transfer can be applied to prepare different highly-oriented polymer materials, especially for conjugated polymers, that exhibit high in-plane anisotropic properties [53,54].

In general, the comprehensive crystal structures and phase transition behavior of PLLA can be strongly influenced by an external field, such as draw temperature, draw rate, and draw ratio [55,56]. Therefore, understanding and controlling the structural formation and orientation play a key role in regulating or enhancing the mechanical performances of polymers. Apart from other methods, such as a normal drawing rate about 5–10 mm/s, the friction transfer speed can even reach a much faster rate of 1 m/s in our present case, which may pave an effective way to controlling the structure and obtaining highly oriented film [57]. However, the relationship of various PLLA crystal modifications affected by temperature and related phase transition behavior is still lacking. Herein, we prepared a highly orientated PLLA thin film by friction transfer technique. By combining the polarized IR and WAXD data, the crystal structure of thin film can be identified at different friction transfer temperatures, and the subsequent phase transition behavior in the heating process as well as acetone treatment has been discussed in detail.

## 2. Experimental Section

### 2.1. Sample

The PLLA 2002D (M_w_ = 2.12 × 10^5^ g/mol, D content of 4.25%) used in the present study was purchased from Natureworks company, Bangkok, Thailand. The PLLA sticks were prepared by injection molding with a barrel temperature of 190 °C for 10 min and were subsequently quenched in cold water to obtain an amorphous state. The gauge length, width, and thickness of the injection-molded test pieces were 80, 10, and 2 mm, respectively.

### 2.2. Characterization

#### 2.2.1. Polarized Optical Microscopy

Polarized optical microscopy (POM) images were obtained using a Nikon H600L polarized optical micro-scope equipped with a MD50 CMOS camera.

#### 2.2.2. Scanning Electron Microscopy

Scanning electron microscope (SEM) analysis was conducted on a JEOL JEM 6700 instrument. Before measurement, the surfaces of the dried specimens were sputtered with gold for SEM observation.

#### 2.2.3. Differential Scanning Calorimetry

Differential scanning calorimetry (DSC) measurement was performed at a heating rate of 10 °C·min^−1^ using a TA Q20 calorimeter under flowing nitrogen.

#### 2.2.4. Fourier Transform Infrared Spectroscopy

Fourier transform infrared spectroscopy (FTIR) spectra were measured using a Bruker tensor 27 spectrometer equipped with a DTGs detector. FTIR spectra were measured at a 2 cm^−1^ spectral resolution, and a total of 32 scans were accumulated in the wavenumber range from 4000 to 450 cm^−1^. The normal transmission mode was employed for polarized IR measurement.

#### 2.2.5. Synchrotron Radiation 2-Dimensional-Wide Angle X-Ray Diffraction (Sr-2D WAXD)

The 2D-WAXD diffraction patterns were collected on beamline (BL16B1) in the Shanghai Synchrotron Radiation Facility (Shanghai, China). The wavelength of the X-ray radiation was 1 Å, and the sample-to-detector distance was 125 mm. An exposure time of X-ray measurement was about 30 s per shot, utilizing a Mar 165 CCD. The temperature dependence of 2D-WAXD measurement was controlled by a Linkam heater with an accuracy of ±0.1 °C, and the heating rate was ca. 2 °C/min.

## 3. Results and Discussion

The friction transfer process of the thus-obtained transparent PLLA stick on the heated glass slide was carried out by a homemade mini friction transfer apparatus, as shown in Figure 1a. The PLLA sample was fixed into the clamps and the screw was tightened to apply the appropriate pressure, while the value of the pressure could be recorded through the pressure sensor. The applied loads chosen here for squeezing and cylinder pressure were 20 kgf/cm^2^ and 0.7 MPa, respectively. The friction transfer speed was about 1 m/s. Highly oriented PLLA films can be obtained in a wide temperature range (T_g_ < T ≤ T_m_); here, the substrate temperatures were set from 70 °C to 150 °C. Before trigging the valve, the amorphous PLLA stick was kept on the glass slide for 5 min at presetting temperature. The friction transfer film, with a thickness of 5 μm, was peeled off in water.

### 3.1. Temperature Dependence of Friction Transferred PLLA Polymorphous Crystal Structure

As shown in Figure 1b, the highly oriented fibril structure of the PLLA chain aligned along the friction transfer direction was generated in the friction transfer process, which was verified by POM (Figure 1c) and SEM (Figure 1d) images. Figure 2 displays 2D-WAXD patterns and 1D-WAXD curves by subtracting the amorphous phase obtained from melt as a function of different friction temperatures in the range of 70 to 150 °C. Diffraction spots can be observed in the 2D-WAXD patterns for the samples friction transferred at a temperature below 150 °C, indicative of highly-oriented structure formation. Fast friction transfer speed can prevent the relaxation of molecular chains, even at a friction temperature near the melting point of PLLA. Due to the complicated peak overlapped of various crystal forms, integrated intensities in the q range from 11 to 13 nm^−1^ of strong characteristic reflections corresponding to each crystalline form among α′-, α-, and β-forms were evaluated carefully by the multiple Gaussian peaks deconvolution method, as shown in Figure 3a. Various crystal structures are remarkably influenced by the friction temperatures in this region. In the peak deconvolution process, three different peak positions representative of α′(100/200), α(100/200), and β(200) were fixed at q = 0.5407, 0.5333, and 0.5205 nm^−1^, respectively [15,28,29].

In the meantime, IR spectra directly reflect the local structure of the single chain; it is very sensitive to the conformation and local molecular arrangement. All the samples at different friction temperatures show the 912 and 921 cm^−1^ characteristic peaks by decomposing overlapped curves, as depicted in Figure 3b. The characteristic IR band of 921 cm^−1^ was assigned to the coupling of the C-C backbone stretching with the CH_3_ rocking mode, which is ascribed to the α- and α′-forms [32,33,55]. Moreover, it was reported that the β-form of PLLA homocrystal had essentially the same 3/1 helical conformation and showed the characteristic band at 912 cm^−1^, as described in the previous literature [30].

In order to quantitatively analyze the molar fraction of each crystal form, the integrated intensities of the α′(110/200), α(110/200), and β(200) reflections were applied for calculating the relative molar fraction of each crystal form. The relative molar content of form f_x_, in which x represents α′-, α-, or β-form individually, was estimated as below.
f_total_ = f_β(200)_ + f_α(200⁄100)_ + f_α’ (200⁄100)_ = 1(1)
f_x_ = I_x_/((I_β(200)_ + I_α(200/110)_ + I_α’ (200/100)_)(2)

Three friction transfer temperature regions of various crystal forms are identified in Figure 4. At low friction temperatures (70–80 °C), only α′ and β crystal forms can be observed with approximately approaching molar content. With increasing friction transfer temperature, the α-form started to generate, and thus three crystal forms including α′-, α, and β-forms were coexistent in the temperature range from 90 to 100 °C. At high friction transfer temperatures (110–140 °C), α′-form cannot be observed. However, the molar content of α-form continued to increase rapidly, while the content of β-form decreased. It should be noticed that β-form exists in all friction transfer temperature regions from 70 to 140 °C, indicating that fast friction transfer rate is conducive to the formation of metastable β-form. To further confirm the molar fraction of β-form by using IR spectra, the relative molar fraction of β-form can be estimated by utilizing characteristic IR bands of 921 and 912 cm^−1^. Substantially congruent with the X-ray diffraction data, the relative content of β-form shows a similar tendency with various friction transfer temperatures, as shown in Figure 5. The relative content of β-form varies from 40 to 60% in the friction transfer temperature region between 70 and 130 °C, whereas the content of β-form drops obviously down to less than 30% at a friction transfer temperature of 140 °C.

### 3.2. Thermal-Induced Phase Transition of PLLA Friction Transferred at 100 °C

In order to disclose the complicated phase transition among these crystal forms in the heating process, the PLLA thin film friction transferred at 100 °C was utilized as an example. Figure 6 shows the DSC curves. It is clear that almost no glass transition can be observed in the first heating procedure indicative of the high crystallinity of this sample. Moreover, two endothermic peaks at 148 and 154 °C can be observed, which should be ascribed to the melting of β- and α-form crystals, respectively. For the second heating, obvious T_g_ at about 62 °C and one melting peak ascribing to α-form can be recognized. Unfortunately, DSC thermogram does not show the procedure of α′-form recrystallization to α-form. For further detailed analysis of phase transition behavior, the phase transition among β-, α′-, and α-forms in the heating process is traced by in situ WAXD measurements in Figure 7a. In fact, there was almost no change in the 1D-WAXD curve in the heating process. In order to emphasize the phase change in the region of high temperature, the relative content of the crystal forms in the heating procedure from 130 to 160 °C is shown in Figure 7b. When the temperature reached 148 °C in the heating process, the content of β-form started to decrease, which corresponded to the melting point of β-form shown in the DSC curves. In the meantime, α-form only increased in the fraction of content, which may suggest the metastable β-form partially changes to a stable α-form. By further heating until 152 °C, the relative content of α′-form began to decrease, while α-form continues to ramp up, which was ascribed to the melting-recrystallization and crystal perfection process of α′-form to α-form phase transition reported in our previous study [58].

### 3.3. Acetone-Induced Phase Transition of PLLA Friction Transferred at 100 °C

Several studies on solvent-induced crystallization behavior have been reported [59,60]. There is, however, as far as we know, no report about the influence of solvent on the phase transition behavior of PLLA, especially for the β-form. As reported, acetone was an effective solvent to accelerate the crystallization of PLLA, which was adopted here to check the influence of it on the morphology and crystal structure of the friction transferred PLLA film.

Figure 8 shows the polarized optical micrographs of the friction transferred PLLA films. The as-prepared films (Figure 8a,c) exhibit smooth surfaces and highly oriented fibril morphology. After acetone immersion for 24 h, as shown in Figure 8b,d, the clearer fibril morphology of PLLA can be observed without losing highly oriented structure. The change of crystal structure was investigated by utilizing WAXD and polarized FTIR techniques. The WAXD results of PLLA film friction transferred at 100 °C before and after immersing into acetone is shown in Figure 9a. Three crystal modifications (β-,α′-, and α-forms) changed to the characteristic peak of pure α-form, which is indicative of acetone inducing the occurrence of phase transition from metastable α′- and β-forms into the stable α-form. This was also confirmed by FTIR measurements, as shown in Figure 9b. It was clear that the coexistence of 912 and 921 cm^−1^ IR characteristic bands changed to only one IR band of 921 cm^−1^ after acetone treatment.

Figure 10a,b show the polarized FTIR spectra of friction transferred PLLA film before and after acetone treatment in the region of 1800–1000 cm^−1^(C=O stretching vibration, CH_3_ and CH bending, and C-O-C stretching region) and 1000–600 cm^−1^(C-C backbone vibrations), respectively [61]. In addition, the band assignments of α′- or α- and β-forms are summarized in Table 1. The film exhibited a strong dichroism at the maximum absorption band, whereas the dichroic ratio D =A///A⊥ for 912 cm^−1^ (β-form) and 921 cm^−1^ (α′- or α-form) was nearly zero. The result demonstrated that the polymer backbones or chain orientation almost perfectly parallel the friction transfer direction. It can be clearly seen that the ordered chain orientation remained parallel to the drawing direction in the strain-induced mesomorphic ordered structure, where the strain-induced 10/3 helical chain started to occur, as discussed previously by the appearance of the characteristic band at 921 cm^−1^.

## 4. Conclusions

In the present study, highly oriented thin films of PLLA have been prepared by a friction transfer method at different temperatures. The polymorphism of the resultant PLLA films and related phase transformation behavior during the heating process and solvent treatment were studied by utilizing WAXD, FTIR, and POM techniques. Preference of β-form was generated between 70 and 140 °C, and the relatively fraction of β-, α′, and α-forms was also evaluated quantitatively. The results suggest that high friction transfer speed was in favor of the formation of metastable β-form, regardless of the friction temperature region from T_g_ to T_m_. The comprehensive thermal-induced phase transformation process among β-, α′-, and α-forms was revealed by in situ WAXD measurement. The contents of β- and α′-forms decreased continuously in the heating process, which may experience partial melt-recrystallization or solid-to-solid transition to α-form. By immersing into acetone solvent, the highly oriented structure was retained, and dichroic ratio D approached 0, whereas the crystal structure of β- and α′-forms transformed to a stable α-form after solvent treatment by keeping the chain orientation. Current research may provide an effective approach to fabricating highly oriented thin film combined by regulating the crystal structure.

## Figures and Tables

**Figure 1 polymers-14-05300-f001:**
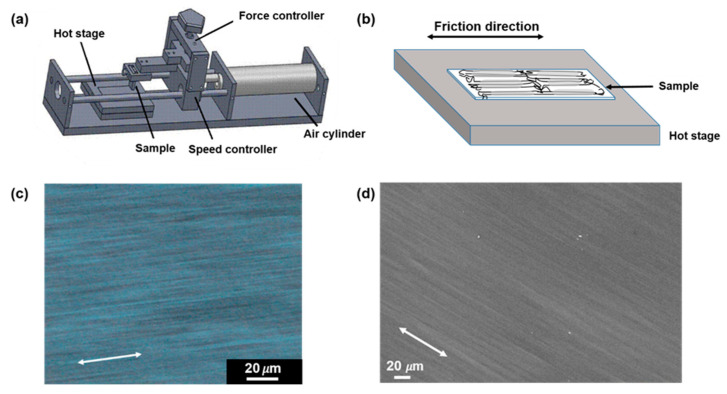
Schematic drawing of homemade mini friction transfer machine (**a**); enlarged region for the PLLA chain axis on the glass slide along the friction direction (**b**); POM (**c**) and SEM (**d**) images of the PLLA films friction transferred at a temperature of 100 °C, respectively. The arrows indicate the friction direction.

**Figure 2 polymers-14-05300-f002:**
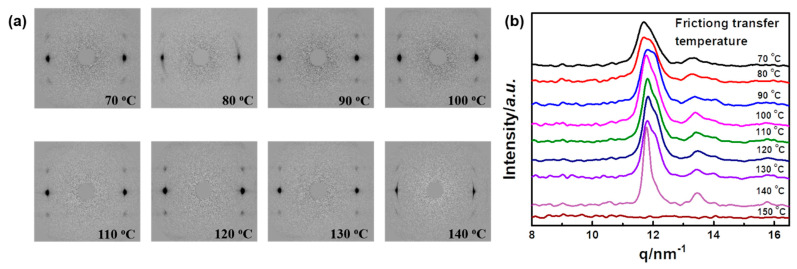
2D X-ray diffraction diagrams (**a**) and corresponding 1D X-ray diffraction profiles (**b**) of friction transferred PLLA at different temperatures. The 2D WAXD patterns processed by subtracting of amorphous part.

**Figure 3 polymers-14-05300-f003:**
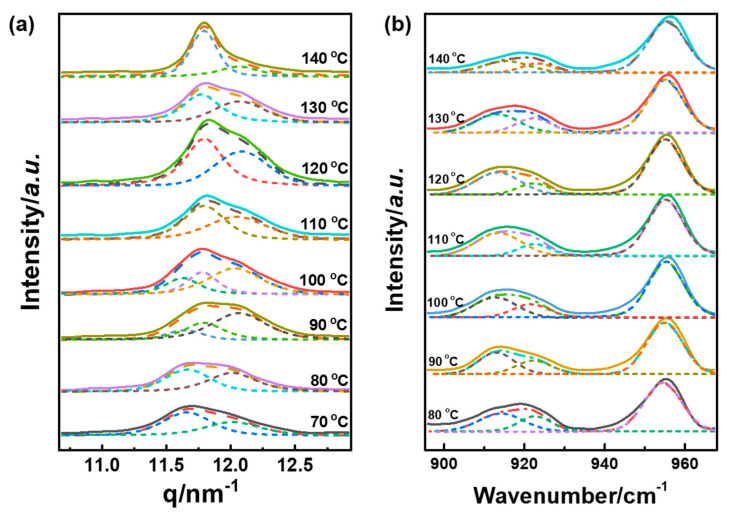
Deconvolution 1D X-ray diffraction profiles (**a**); and FTIR spectra (**b**); (dot line) of PLLA at different friction transfer temperatures. The solid lines are the experimental spectra, the dotted lines are the fitting result.

**Figure 4 polymers-14-05300-f004:**
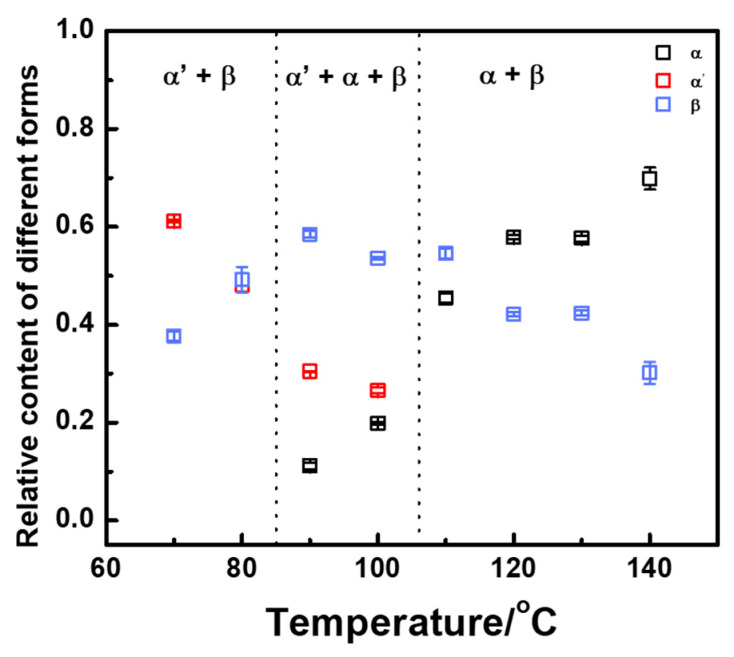
Relative content of different crystal forms versus friction transfer temperature extracted from 1D-WAXD profiles.

**Figure 5 polymers-14-05300-f005:**
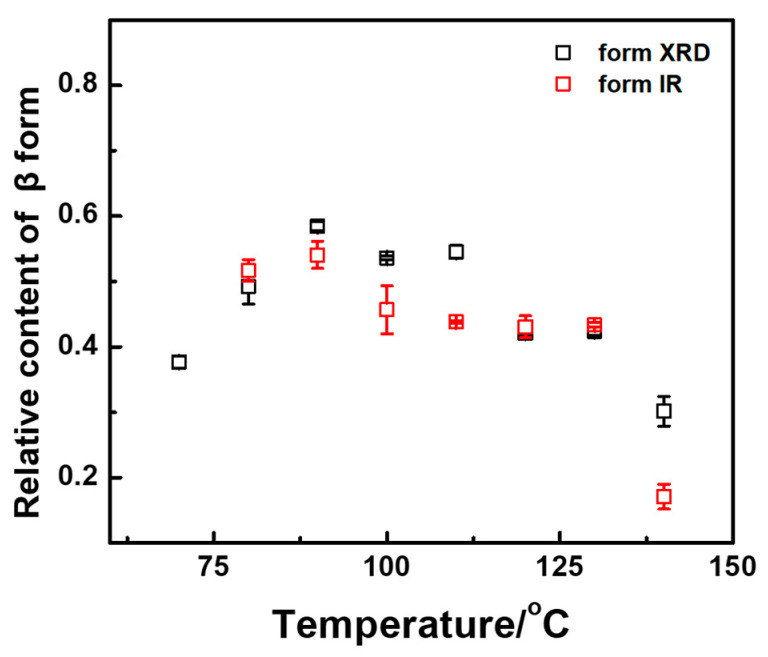
Comparison of relative content of β-forms versus friction transfer temperature extracted from 1D-WAXD profiles and FTIR spectra, respectively.

**Figure 6 polymers-14-05300-f006:**
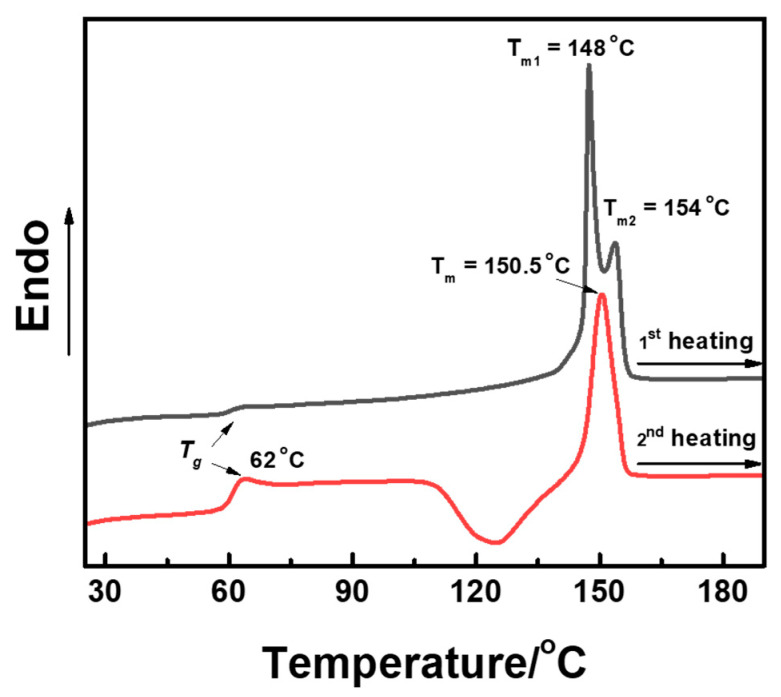
The DSC heating curves of oriented PLLA film friction transferred at 100 °C recorded in the first and second runs with heating rate of 10 °C/min.

**Figure 7 polymers-14-05300-f007:**
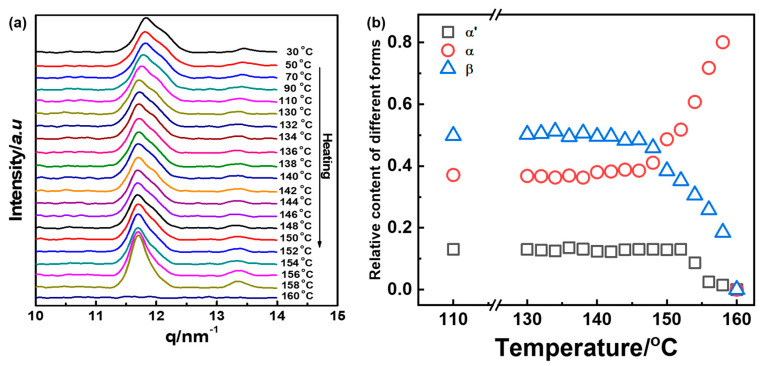
(**a**) Temperature-dependent WAXD curves of oriented PLLA film friction transferred at 100 °C. (**b**) change of relative content of α′-, α-, and β-forms in the heating process. The heating rate is 5 °C/min.

**Figure 8 polymers-14-05300-f008:**
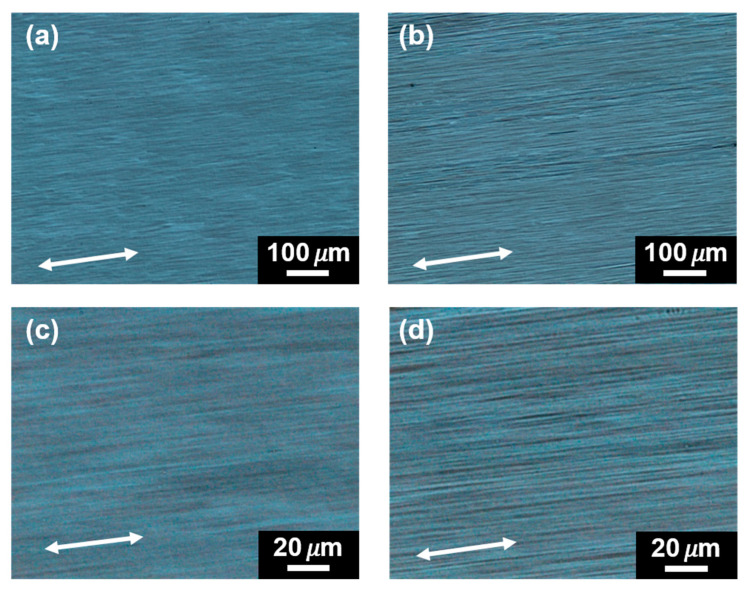
Polarized optical micrographs of a PLLA film friction transferred at 100 °C before (**a**,**c**) and after (**b**,**d**) being immersed in acetone solvent. The friction transfer direction is indicated by an arrow.

**Figure 9 polymers-14-05300-f009:**
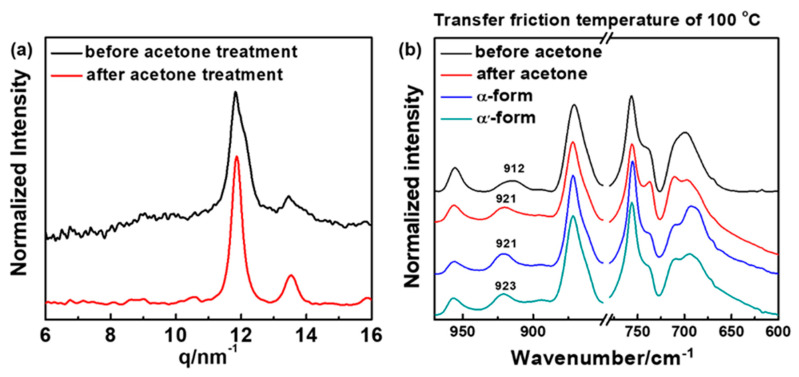
(**a**) 1D X-ray profiles and (**b**) FTIR spectra of oriented PLLA films friction transferred at 100 °C before and after acetone treatment for 24 h. For comparison, FTIR spectra of PLLA α-form annealing at 125 °C for 2 h and α′-form annealing at 90 °C for 2 h in the region of 1000-600 cm^−1^ are shown here.

**Figure 10 polymers-14-05300-f010:**
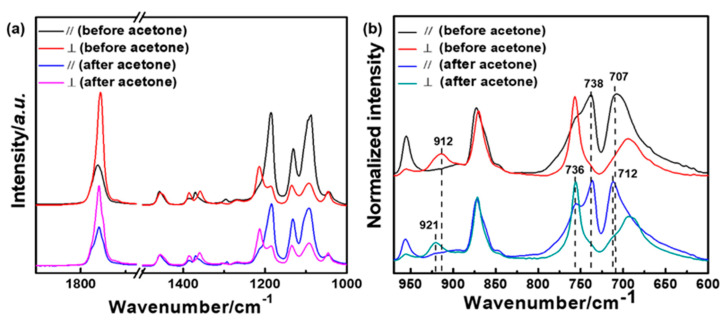
Polarized FTIR spectra of oriented PLLA film friction transferred at 100 °C before and after acetone treatment for 24 h in the wavenumber region of 1800-1000 cm^−1^ (**a**) and 1000-600 cm^−1^ (**b**), respectively.

**Table 1 polymers-14-05300-t001:** Characteristic infrared band assignments of PLLA in the range of 1800-600 cm^−1^.

IR Frequencies (cm^−1^)				Polarization	Assignments
Amorphous	α’	α	β		
	1761	1759		⊥	ν(C=O)
	1457	1457			δ_as_(CH_3_)
	1386			⊥	δ_s_(CH_3_)
		1370		//	δ_s_(CH_3_) + δ(CH)
	1360	1360		⊥	
1213		1213		/	ν_as_(COC) + r_as_(CH_3_)
1183	1183			//	
1131	1134			//	r_as_(CH_3_)
1092	1092	1092		//	ν_s_(COC)
1046	1046	1045			ν(C-CH_3_)
955	957	957		//	r(CH_3_) + ν(C-COO)
	921	921	912	⊥	ν(C-C) + r(CH_3_)
870	872	872			ν(C-COO)
757	757	757		⊥	δ(C=O)

## Data Availability

The data will be available on demand.

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
