# Peer review of "Temperature-Dependent Polymorphism and Phase Transformation of Friction Transferred PLLA Thin Films"

_polymers, 2022, doi:10.3390/polym14235300_

Round 1

Reviewer 1 Report

The manuscript is interesting and presents useful results. However, there are a few concerns that MUST be addressed:

1. Abbreviations in the abstract section should be introduced first prior to their use.

2. What was the basis for selecting the squeezing load and friction speed? Please elaborate.

3. Line 126 states that "highly oriented fibril structure......." The results should be compared with the available literature and should demonstrate the importance of the results.

4. While relating the characteristic reflection. Please cite the appropriate references.

5. Similarly the relationship between the peaks and corresponding bands of IR spectra should be supported by suitable references.

6. The author states that "When the temperature reached 148 C the content of .............." state the reasons. Further, illustrate behavior is different and compare to the literature.

Author Response

Response to Reviewer 1 Comments

The manuscript is interesting and presents useful results. However, there are a few concerns that MUST be addressed:

Authors’ response: We thank the reviewer for the positive comments on our manuscript.

Point 1: Abbreviations in the abstract section should be introduced first prior to their use.

Response 1:

According to review’s suggestion, we have corrected and introduced the full names and abbreviations of wide-angle X-ray diffraction (WAXD), Fourier transform infrared spectroscopy (FTIR) and polarized optical microscopy (POM) measurements in Abstract part. And also Poly(D-lactic acid) (PDLA) has been corrected in Introduction part.

Point 2: What was the basis for selecting the squeezing load and friction speed? Please elaborate

Response 2:

Thanks for reviewer’s comment. More details of friction transfer process have been added and refined. The PLLA sample was fixed into the clamps and screw was tightened up to apply appreciate pressure, whereas the value of pressure could be recorded through the pressure sensor. The applied loads chosen here for squeezing and cylinder pressure were 20 kgf/cm2 and 0.7 MPa, respectively. The friction transfer speed was about 1 m/s. In fact, squeezing load and friction speed were inversely proportional. The greater squeezing load, the slower friction speed. In our case, we selected suitable squeezing pressure and cylinder pressure to ensure that we could obtain a thin film with high orientation.

Point 3: Line 126 states that "highly oriented fibril structure......." The results should be compared with the available literature and should demonstrate the importance of the results.

Response 3:

In the introduction part, we emphasized that the drawing rate for normal or traditional method was about 5-10 mm/s, much slower than friction transfer method. In our results, orientation parameters of friction transfer method at different temperatures were about between 0.95 and 0.97 estimated by 2D-WAXD data. Compared to other literatures like solution-spun[34], solid-state coextrusion[35], under pressure and shear[36], this approach can offer a simple and versatile method for ordering a wide variety of crystalline materials. In addition, high friction speed can decrease orientation time to prevent the chain relaxation on the substrate. Because near the melting temperature, slow friction or shear speed may cause the molten or inhomogeous distribution of the film. We added the sentence “Fast friction transfer speed can prevent the relaxation of molecular chain even at the friction temperature near melting point of PLLA” in the paper.

Point 4: While relating the characteristic reflection. Please cite the appropriate references.

Response 4:  

According to review’s suggestion, we cited suitable references for explaination of characteristic crystal planes and marked in this paper.

Point 5: Similarly the relationship between the peaks and corresponding bands of IR spectra should be supported by suitable references.

Response 5:

According to review’s suggestion, we cited suitable references of IR band assignment and marked in this paper.

Point 6: The author states that "When the temperature reached 148 C the content of .............." state the reasons. Further, illustrate behavior is different and compare to the literature.

Response 6:

We made a mistake about the sample information for molecular weight. In this study, we used the commecial PLLA 2002D from Natureworks companys. According to the relative literatures (Express Polym. Lett. 2011, 5, 82–91; Polym. Degrad. Stab. 2010, 95, 116e125), it showed the melting temperature about 150 oC, which was consist with our results. Because it had different D-lactic monomer content of 4.25 and weight average molecular weights was 212 kDa respectively, assessed by Gel Permeation Chromatography (GPC). Therefore, the relative low molecular weight and high D content lead to the lower melting temperature compared to normal PLLA samples. Therefore, the information of PLLA sample has been corrected in the paper.

In addition, we seleted the sample including β-, α′- and α-forms, as the literature[36] and [39] reported, the β form partially transformed to α form through melt-recrystallization process. And also α′-form may experience crystal perfection process during phase transition as reported in literature [58], our results showed similar tendency are consist with the previous studies.

Reviewer 2 Report

Authors present results on polymorphism and phase behavior as observed from extensive experimental studies on PLLA thin films. The work is interesting, the experimentation is complete, and the presentation of the results is convincing. As I do not have any major comments subject to addressing the following minor ones the work would be publishable in Polymers.

) In Fig. 6 DCS curves for 148oC show a small -but non-zero- change near the 60 oC. However, authors in the text and in the graph identify T_g only at 154oC.  

) Fig. 3: it is not easy to distinguish the raw experimental curves (spectra) from the deconvoluted profiles. For this, authors could select other formats for curve representation (maybe profiles after the deconvolution?)

) Indices in Equations 1 and 2 are not consistent with the ones in text lines 156-158.

) Error bars should be added when possible, especially in Figures 4, 5 and 7(b). In addition, authors should identify in the legend of Fig. 4 the vertical dotted lines as indicating the transition between the three different regimes of crystal forms.  

) The manuscript requires a very careful editing for grammar and syntax errors.

. Line 17: “were estimated” -> “was estimated”

. Lines 41 and 165: “and generates” -> “and is generated”

. Line 118: “was set” -> “were set”

. Line 151: “by substract” -> “by substracting”

. Sentences in lines 141-142 and 163-164 are not clear and should be rephrased.

Author Response

Response to Reviewer 2 Comments

Authors present results on polymorphism and phase behavior as observed from extensive experimental studies on PLLA thin films. The work is interesting, the experimentation is complete, and the presentation of the results is convincing. As I do not have any major comments subject to addressing the following minor ones the work would be publishable in Polymers.

Authors’ response: We thank the reviewer very much for the positive comments and we are also glad that the reviewer found some interesting results in our manuscript

Point 1: ) In Fig. 6 DCS curves for 148oC show a small -but non-zero- change near the 60 oC. However, authors in the text and in the graph identify Tg only at 154oC

Response 1:

Thanks for reviewer’s suggestion. In the first heating curve, because the film was obstained by friction transfer method with high crystallinity, it showed no clear Tg. In the second heating curve, it showed obvious Tg due to low crystallinity during cooling process. In addition, the Tm1 and Tm2 were ascribed to the melting of β- and α-form crystals as we mentioned in the paper. The melting point of Tm=150.5 oC should be the melting of α-form,which was generated in the cooling process from molten state. And also the values of Tg and melting point were added in Figure 6.

Point 2: Fig. 3: it is not easy to distinguish the raw experimental curves (spectra) from the deconvoluted profiles. For this, authors could select other formats for curve representation (maybe profiles after the deconvolution?)

Response 2:

Thanks for reviewer’s suggestion. We've refined Fig.3 even further to make it easy to distinguish the raw experimental curves from the deconvoluted profiles. In Fig.3, the solid lines represent the raw experimental spectra, and the dotted lines represent the fitting result. We have also made corrections in this paper.

Point 3: Indices in Equations 1 and 2 are not consistent with the ones in text lines 156-158.

Response 3:

We have corrected the mistake and exchanged the order of equations.

Point 4: Error bars should be added when possible, especially in Figures 4, 5 and 7(b). In addition, authors should identify in the legend of Fig. 4 the vertical dotted lines as indicating the transition between the three different regimes of crystal forms. 

Response 4:

Thanks for reviewer’s suggestion. We take into account the error of fitting and changed figure 4 and figure 5 in this paper. In the legend of Fig.4, the vertical dotted lines are not strict boundary to distinguish between three areas because of the interval of temperature is about 10 oC. The rough boundary was shown here just to indicate the three regions for the formation of different crystal forms.

Point 5: The manuscript requires a very careful editing for grammar and syntax errors.

. Line 17: “were estimated” -> “was estimated”

. Lines 41 and 165: “and generates” -> “and is generated”

. Line 118: “was set” -> “were set”

. Line 151: “by substract” -> “by substracting”

. Sentences in lines 141-142 and 163-164 are not clear and should be rephrased.

Response 5:

According to the reviewer’s suggestion, we have tried our best to correct all the mistakes in revised version and had the manuscript polished with a professional assistance in writing.

Round 2

Reviewer 1 Report

The manuscript is revised satisfactorily.